# Life Cycle Assessment of Thermoelectric Generators (TEGs) in an Automobile Application



**Kotaro Kawajiri [1],*, Yusuke Kishita [2] and Yoshikazu Shinohara [3]**

1    National Institute of Advanced Industrial Science and Technology, Tsukuba 305-8569, Japan
2    School of Engineering, The University of Tokyo, Tokyo 113-8656, Japan; kishita@pe.t.u-tokyo.ac.jp
3    Center for Green Research on Energy and Environmental Materials, National Institute for Materials Science, Tsukuba 305-0047, Japan; SHINOHARA.Yoshikazu@nims.go.jp
*    Correspondence: kotaro-kawajiri@aist.go.jp; Tel.: +81-29-861-8089

**Abstract:** In this paper, a possibility to reduce the environmental burdens by employing thermoelectric generators (TEGs) was analyzed with a cradle-to-grave LCA approach. An upscaling technique was newly introduced to assess the environmental impacts of TEGs over its life cycle. In addition to $CO_2$ emissions, other environmental impacts as well as social impacts were assessed using the Life Cycle Impact Assessment Method based on Endpoint Modeling (LIME2). The analysis was conducted under two scenarios, a baseline scenario with a 7.2% conversion efficiency and a technology innovation scenario with that of 17.7% at different production scales. The results showed that while GHG emissions were positive over the life cycle under the baseline scenario, it became negative ($-1.56 \times 10^2$ kg-$CO_2$ eq/kg) under the technology innovation scenario due to GHG credits in the use phase. An increase in the conversion efficiency of the TEG and a decrease in the amount of stainless steel used in TEG construction are both necessary in order to reduce the environmental impacts associated with TEG manufacture and use. In addition, to accurately assess the benefit of TEG deployment, the lifetime driving distance needs to be analyzed together with the conversion efficiency.

**Keywords:** emerging technology; life cycle assessment; scaling effect; thermoelectric generator

## 1. Introduction

The use of fossil fuels to generate electricity has had a variety of negative impacts on our society, primarily in the form of atmospheric pollution and global warming. The global consumption of oil increased from 95 million barrels per day in 2014 to 100 million barrels in 2018 [1]. The cost of electricity has been increasing in recent years due to the limited supply of oil and economic and political factors [2]. Continuation of using fossil fuels will soon reach a critical point from a sustainability point of view [3]. Therefore, using carbon-free renewable energy, which can be retrieved from sources such as RF (Radio Frequency) radiation, geothermal, solar, and other natural sources, and converting this energy into electricity has attracted considerable attention [1]. Among them, thermal energy is abundantly available and has numerous sources, including electronic devices (phones, computers, and other electronic devices), automobiles, buildings, and air conditioners.

According to the International Organization of Motor Vehicle Manufacturers (OICA), road transport is responsible for approximately 16% of global carbon dioxide ($CO_2$) emissions [4]. Numerous automotive manufacturers are focusing on finding alternative power sources to reduce fuel energy costs and GHG emissions. The amount of heat emitted from automobiles is very high and can range from 100 to 800 °C with a thermal power of up to 10 kW [5]. This amount of heat, which would typically be wasted, could be a remarkable source of sustainable energy. Thus, converting this wasted heat into energy brings us from a linear to circular economy from economical as well as environmental aspects.

One of the ways in which GHG emissions can be reduced in automobiles is to use a thermoelectric generator (TEG) [6]. A TEG is a device that is used to convert heat into

electricity, and TEG devices are considered to be promising for achieving a low-carbon society as they can be used to generate electrical energy from waste heat. However, as of 2015, the widespread application of TEGs for power generation is limited by the relatively low conversion efficiency of existing TEG systems (i.e., 5–10%) [7]. In order for TEGs to be more widely adopted, it is imperative that their conversion efficiency be improved. In addition to the technological issues mentioned above, it is also necessary to evaluate the environmental burdens associated with the use of TEGs in real applications.

However, relatively few studies on the environmental effects of TEGs have been conducted to date [6]. In addition, the low energy efficiency of TEGs and their high manu-facturing costs means that they are currently not cost effective to produce [8,9]. Several life cycle assessment (LCA) studies have examined TEG systems. For example, Karvonen et al. published a review of published literature and examined patents for waste-heat recovery technology, including thermoelectric generation [10]. Evangelisti et al. conducted an LCA of TEG systems in the treatment of municipal solid waste [11]. Murphy et al. conducted an LCA study on the Irish wood processing industry [12]. Kishita et al. recently assessed the future scenarios of GHG emissions from TEG systems for passenger automobiles [6].

By literature review on the previous studies, it was identified that three issues remain to be resolved: (1) few studies have conducted cradle-to-grave LCA analyses on TEG systems, (2) none of those studies considered the effect of scale (e.g., upscaling) on a commercial scale from the lab scale, and (3) the effect of TEGs on other environmental burdens and social impacts were not addressed.

Cradle-to-grave LCA analysis is necessary to correctly assess the merit/demerit of TEGs, as use and EOL phase take key roles in an assessment of the environmental burdens. As for an assessment of a mass production scale, without the upscaling effect from lab to mass production scale in consideration, GHG emissions induced by the manufacture of TEG devices were likely overestimated at the mass production scale. In addition, other environmental and social impacts associated with the use of TEGs have not yet been clarified. However, while TEG usage may reduce GHG emissions once deployed, potentially significant resource consumption problems may be associated with the production of these systems as they typically contain rare metals, such as selenium (Se), tellurium (Te), bismuth (Bi), and antimony (Sb). Clearly, this aspect of TEG manufacturing needs to be considered through a cradle-to-grave analysis in order to comprehensively understand the benefits of using TEG systems before TEGs can be considered for widespread application. It is thus very important to address these issues before implementing TEGs in real-world applications.

It is common practice to evaluate a future technology at a research scale before mass production. While it is advantageous to evaluate an emerging technology in the research and development phase of a product's development, LCA of emerging technologies is an ongoing challenge [13]. Emerging technologies are developed in inefficient small-scale laboratories [14]; however, process efficiency usually increases as the production scale increases [15]. Since LCA requires "apple to apple" comparisons, implementing upscaling analyses becomes necessary in order to assess the GHG emissions induced by mass scale production. Here, the concept of scaling is introduced as a means of assessing the environmental impacts that are induced as production is upscaled from the research scale to the mass production scale.

The objective of this study is to assess the environmental impacts as well as social impacts associated with automotive TEG usage by addressing the aforementioned three issues under two different scenarios: a baseline scenario and a technology innovation scenario. First, a cradle-to-gate analysis of GHG emissions was conducted for TEG assembly. Then, based on the obtained results, analyses of the two scenarios were conducted. As TEGs represent a new technology and are still in the research stage, the concept of upscaling was employed in this study and an LCA approach was used to assess the impacts of TEG systems over the course of their lifetimes. Following Kishita et al. [5], who investigated the GHG emissions and costs induced by the development of TEG systems, this study added the following:

- Cradle-to-grave LCA analysis.
- Assessing the effect of scale (e.g., upscaling).
- Assessing the effect of TEGs on other environmental burdens and social impacts.

By including these three aspects in the analysis, it becomes possible to assess the environmental impact of TEG devices more holistically at a mass production scale. This analysis is considered to be useful for quantitatively assessing the feasibility of adopting TEG systems in an automobile application from environmental as well as social viewpoints. In addition, our results are compared with previous LCA works and where the difference came from and how they should be interpreted is discussed.

## 2. Methodology

### 2.1. Analysis Method

The TEG device considered here is a bismuth-telluride-based (Bi-Te) TEG. The specifications of the Bi-Te TEG are shown in Table 1. The analysis was conducted in three steps.

**Table 1.** Specifications of TEG.

| Item | Value |
| --- | --- |
| Size | $50 \times 50 \times 4.2$ mm |
| Conversion efficiency | 7.2% |
| Maximum allowable temperature | 280 °C |
| Maximum power output | 24 W module$^{-1}$ |
| Total weight of TEG | 47 g·module$^{-1}$ |
| Weight of Bi-Te thermoelectric material in TEG | 27 g·module$^{-1}$ |

1. GHG emissions of the TEG fabrication process were estimated with an upscaling technique from Case 1 to Case 4. The conditions for each case are shown in Table 2.

**Table 2.** Production scales.

| | Unit | Case 1 | Case 2 | Case 3 | Case 4 |
| --- | --- | --- | --- | --- | --- |
| Electric furnace | $\ell$ | 0.011 | 1.000 | 10.000 | 30.000 |
| Ball mill | $\ell$ | 0.011 | 1.000 | 10.000 | 30.000 |
| Hot press | m$^2$ | 0.003 | 0.010 | 0.100 | 1.000 |

2. GHG emissions were assessed based on two scenarios, a baseline scenario and a technology innovation scenario, from Case 1 to 4. Here, a cradle-to-grave analysis was conducted. The baseline scenario (A-1) was assumed to have a conversion efficiency of 7.2%, while the technology innovation scenario was assumed to have a conversion efficiency of 17.7%. Statistics in Suita City, in Japan, about its population, the automobile quantity, and its usage model, such as the distance run per day, were used in this analysis.
3. A cradle-to-grave impact assessment for the environmental and social impacts were conducted using Life-cycle Impact Assessment Method based on Endpoint Modeling (LIME2). The analysis flow is shown in Figure 1.

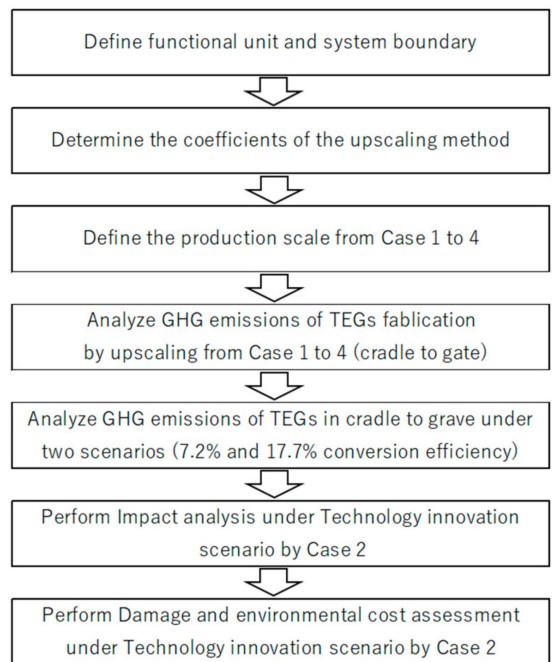

**Figure 1.** Analysis flow.

## 2.2. Functional Units and System Boundaries

The functional unit of this study is one TEG device, as shown in Table 1.

The system boundaries for the TEG heat exchanger and the TEG fabrication processes are shown in Figure 2. For the LCA, the following five stages were considered in TEG production: (1) fabrication, (2) assembly, (3) distribution, (4) use, and (5) end-of-life (EOL) [6]. Since the fabrication and assembly stages consume the most and least amounts of electricity, respectively [6], the upscaling analysis was applied to the fabrication stage in this study. TEG fabrication consists of three processes. First, raw materials such as Te, Se, Bi, and Sb are melted into an ingot in an electric furnace. Then, the ingot is crushed using a ball mill. Finally, the particles are shaped into the desired form using a hot press.

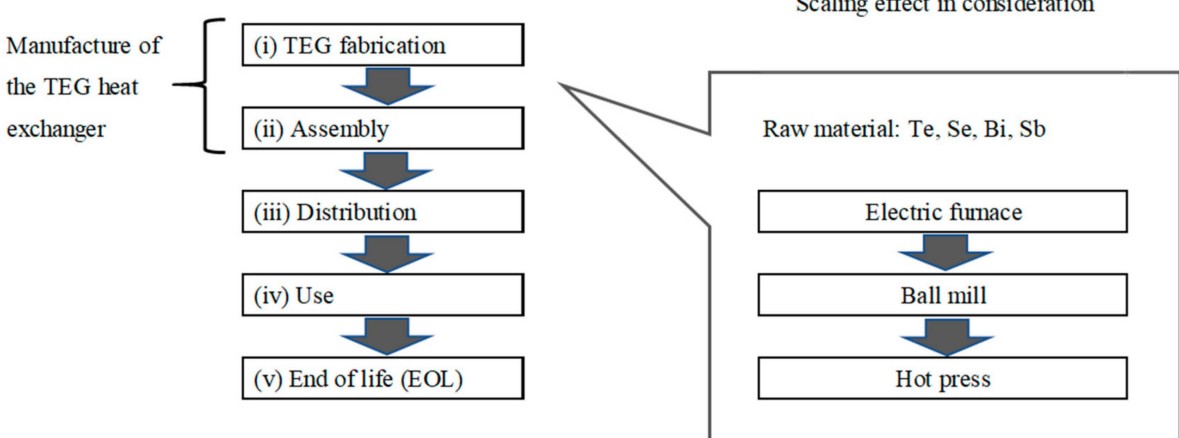

**Figure 2.** System boundaries and TEG fabrication process.

## 2.3. Upscaling Analysis

The inventory data used in this study were extracted from a report by Kishita et al. [6]. The database IDEA_v2.1.3 [16] was used for inventory analysis. We considered the scaling effect on the energy consumed when producing the TEG systems. In addition, information

on the raw materials used in the TEG fabrication process was obtained from an interview with a company that fabricates TEG systems.

To estimate the energy consumption of the fabrication machines at a different scale from that of the lab-scale inventory dataset, an indirect upscaling method [13] was used in this case study. Using this indirect method, the input energy required to produce a given output is expressed as follows:

$$P = P_0 \left( \frac{S_x}{S_0} \right)^f \tag{1}$$

where $P$ represents power, $S$ is the scale of the process, $f$ is the scaling factor, subscript 0 represents the values for the lab scale, and subscript $x$ represents the values for the target scale; the referenced values of scale and power are obtained from Kishita et al. [6]. The scaling factor $f$ is obtained from the specifications of the machines used in the different production scales. The scaling factors for the electric furnace, ball mill, and hot press are shown in Figure 3.

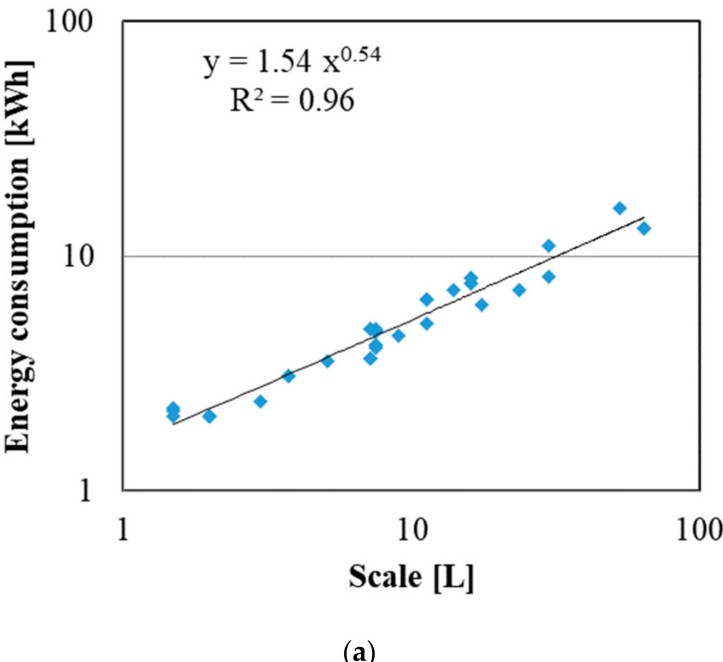

(**a**)

**Figure 3.** *Cont.*

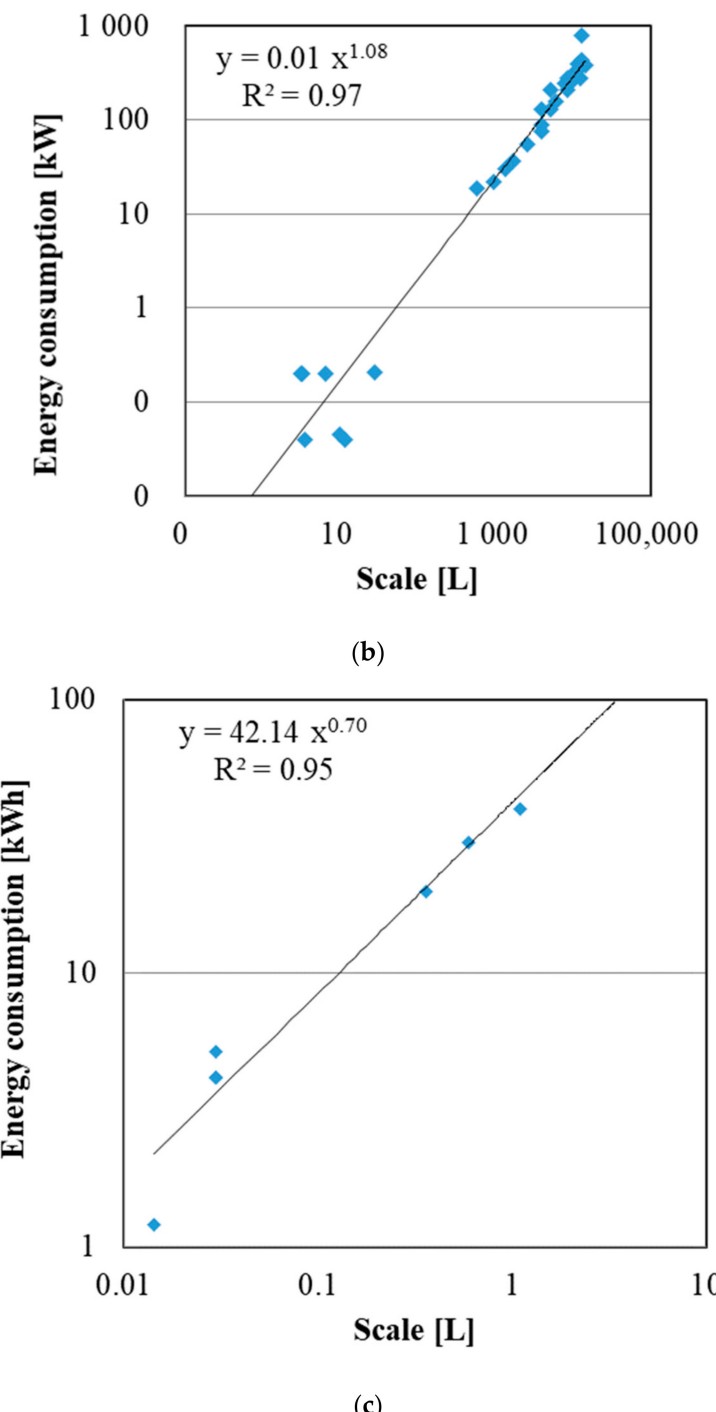

**Figure 3.** Scaling factors for machines. (**a**) Electric furnace; (**b**) ball mill; (**c**) hot press.

The GHG emissions were extrapolated from the research scale (Case 1) to three production scales: Case 2, Case 3, and Case 4. The latter three cases were selected in consultation with an expert. The details of the production scale for each fabrication process are shown in Table 2.

The LIME2 database (JLCA, 2013) [17] in IDEA_v2.1.3 was used for the LCA. Electricity generated by a TEG heat exchanger in the usage phase was treated as a credit and negative GHG values were assigned.

### 2.4. Scenario Analysis

In this study, the environmental impacts associated with the production and use of Bi-Te TEG heat exchangers in passenger cars in Suita City, Osaka, Japan, in 2030, were analyzed. It was assumed that it will be technologically possible to employ TEG heat exchangers in passenger cars by that date [18,19]. In addition, we assume that a typical passenger vehicle has a 12-year lifespan [20] and that the estimated number of gasoline passenger cars is 79,039 in 2012 and 75,780 in 2030 [21,22]. With the collaboration of the local government of Suita City, the annual driving distance [23] was determined and an assessment of the magnitude of gasoline savings associated with TEG adoption was estimated. Kishita et al. [6] analyzed GHG emissions based on four scenarios for TEG conversion efficiency and passenger car location. In this study, a baseline scenario (A-1) and a technology innovation scenario (A-2) were employed using data for passenger cars in Suita. The current conversion efficiency is 7.2% for the A-1 scenario and the advanced conversion efficiency is 17.7% for the A-2 scenario in 2030 [6].

### 2.5. Impact and Social Analysis

Impact assessment as well as the damage and the environmental cost assessments (social assessment) in terms of technology innovation scenario (A-2) were performed based on the LIME2 category.

## 3. Results

The electricity consumption associated with each stage of TEG fabrication in the four cases is shown in Figure 4. As shown in the figure, as the electricity consumption of the electric furnace and hot press are reduced as the production scale increases, the electricity consumption of the ball mill increases. The scale factor for the ball mill is >1, indicating that the electricity consumption increases with scaling. Usually, the scaling effect reduces energy consumption as the capacity of the machine increases. For example, as the capacity of the oven increases so does its volume, but the heat loss as a proportion of the surface area actually decreases as the size of the oven increases. This implies that the scaling effect works in the case of the oven. However, as the capacity (i.e., volume) of the ball mill increases, more energy is required to create the friction generated by the larger and heavier mill; hence, the scaling effect does not work with an increase in the scale of the mill. Consequently, Case 4, which was characterized by a relative decrease in electricity consumption by the electric furnace and hot press compared to the increase in electricity consumption by the ball mill over the production scale, was optimal for the production scale. Further increasing the production scale would increase electricity consumption due to the ball mill and would adversely affect the environment.

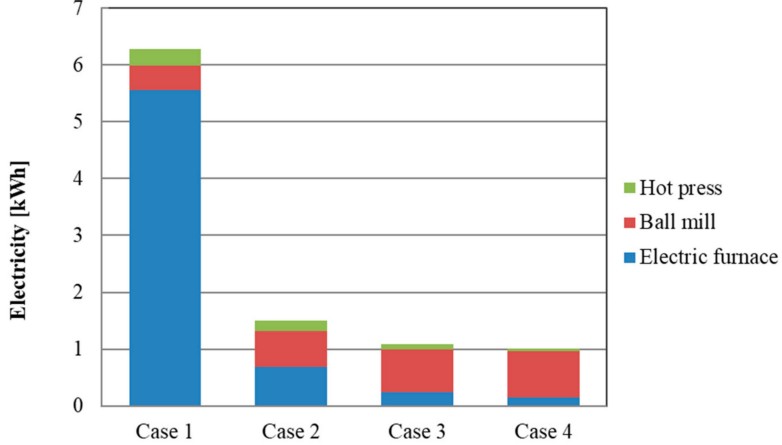

**Figure 4.** Electricity consumption associated with each process under different fabrication scenarios.

GHG emissions in two scenarios are shown in Figure 5. The majority of GHG emissions come from material production. This decreasing trend in GHG emissions is the same as that of electricity consumption shown in Figure 4 because GHG emissions are directly related to electricity consumption. During TEG usage, electricity is produced from waste heat. This electricity is counted as a GHG credit and negative GHG emission values are assigned. Since TEG devices are recycled, zero GHG emissions were assigned to the recycling process. For the baseline scenario, the GHG credit was 97.1 kg-$CO_2$ eq/kg in all four cases. The total GHG emissions are larger than the GHG credit in Cases 1 and 2. However, in the technology innovative scenario with its higher conversion efficiency, the GHG credit was 253.2 kg-$CO_2$ eq/kg, which was larger than the life cycle GHG emissions in all four cases.

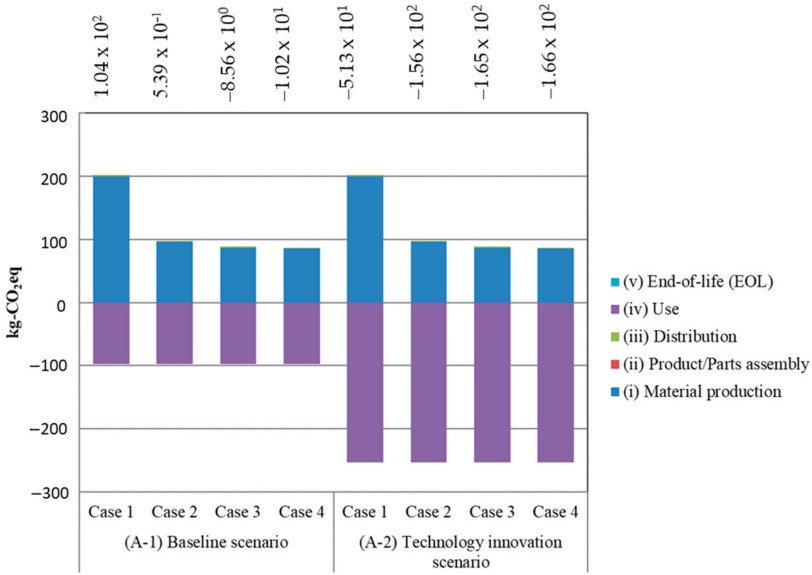

(**a**) GHG emissions from each process.

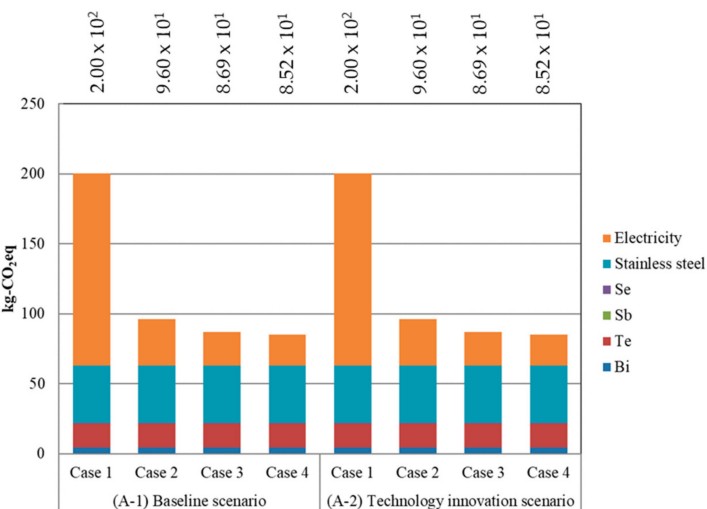

(**b**) GHG emissions from materials.

**Figure 5.** GHG emissions under two production scenarios focusing on (**a**) processes and (**b**) materials used in fabrication.

Regarding GHG emissions by materials, the only contributors to GHG emissions are electricity and raw materials. Electricity comprises the larger portion, accounting for 68%

in Case 1 and 26% in Case 4. Due to the scaling effect, GHG emissions due to electricity decrease from 137.1 kg-CO$_2$ eq/kg in Case 1 to 22.1 kg-CO$_2$ eq/kg in Case 4. Among raw materials, stainless steel occupies the largest portion, as a large amount of stainless steel is used to encapsulate the TEG material to fabricate the heat exchanger. GHG emissions induced under the baseline scenario and under the technology innovation scenario are the same because the exact same raw materials are used to fabricate the TEG. The only difference is that the technology innovation scenario has a higher conversion efficiency. The level of GHG emissions induced by the raw materials remain constant over the production scale as raw material consumption remains constant at a given scale. As a result, stainless steel is the largest contributor to GHG emissions (41.1 out of 85.2 kg-CO$_2$ eq/kg) in Case 4 under both scenarios.

The results of the LCA under the technology innovation scenario are shown in Figure 6. Gasoline is the amount of gasoline saved by the TEG in the passenger vehicle. Diesel corresponds to materials, such as the trucks that are used for distribution in the system boundary. Here, we used Case 2 as the reference because the scale for this case is considered to reflect current conditions [6]. Depending on the category of impacts, there is a possibility that the environmental footprint might increase. In particular, the stainless steel used to fabricate the TEG heat exchanger has a marked impact on most of the impact categories because a large amount of stainless steel is used to encapsulate the TEG. The influence of Te has a marked effect on ozone layer depletion, accounting for 96% of ozone depletion in the atmosphere. In addition, Te also has a deleterious effect on the human toxicity (cancer) category. In the resource consumption category, Bi has the largest impact on the resource consumption category at 88%, implying that alternatives to Bi should be sought in the future.

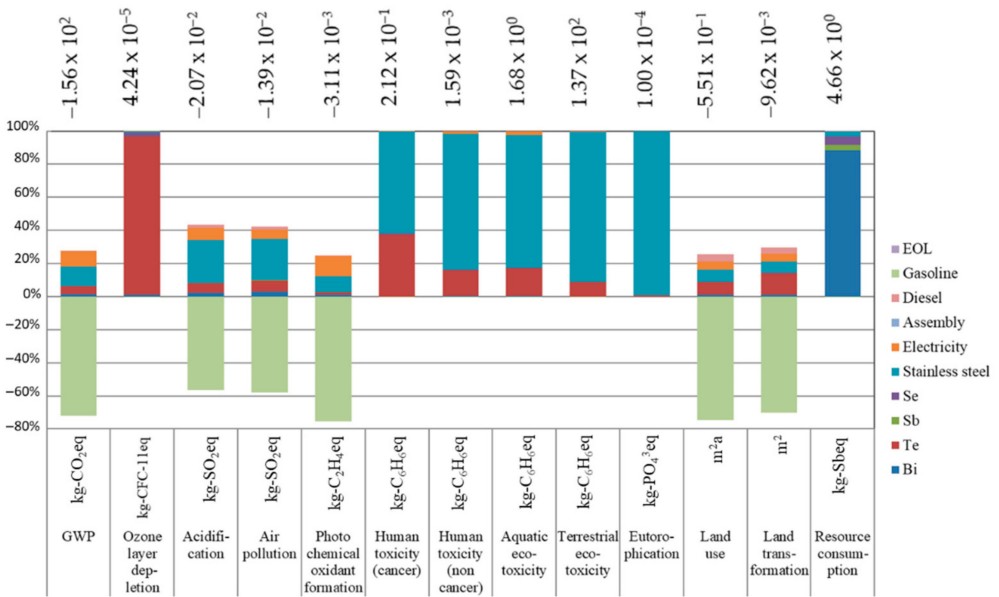

**Figure 6.** Impact assessment for technology innovation scenario (Case 2).

Figure 6 shows that the GHG credit due to the gasoline savings generated by the TEG played a major role in GHG emissions. GHG reduction attributed to the TEG is 253.2 kg-CO$_2$ eq/kg, indicating that total GHG emissions (GWP) are $-1.56 \times 10^2$ kg-CO$_2$ eq/kg. The credit also plays a role in reducing acidification, air pollution, photochemical oxidant formation, land use, and land transformation, resulting in the negative scores.

However, since the coefficient of resource consumption for Te is not included in the LIME2 database, the result obtained for resource consumption would change if the impact of Te is considered.

The results of the damage and the environmental cost assessments (social impacts) estimated under the technology innovation scenario (Case 2) are shown in Figure 7. The

figure shows that the effect of stainless steel is dominant in all of the categories for the same reasons given for Figure 6, as a large amount of stainless steel is used to encapsulate the TEG devices. As shown in Figure 4, since energy consumption does not decrease with an increase in the scale of production, there is little room for reducing the environmental impact through changes in energy consumption. The reduction in gasoline utilization attributed to TEGs contributed to the negative values obtained for the human health and primary production categories. In terms of the environmental costs (single index), the credit was equivalent to 1055 JPY, which meant that the total index was 1466 JPY. Therefore, based on the results, the key factors for reducing the environmental impacts are to increase the efficiency of the TEG heat exchanger and to reduce the amount of stainless steel used in fabrication. In addition, substitution of or reduction in the Te in the TEG would decrease the impacts of this element on human health, biodiversity, and primary production. As stated previously, one limitation of this study is that the coefficient of resource consumption for Te is not included in the LIME2 database. Consequently, the data in the resource consumption category are likely to be underestimated.

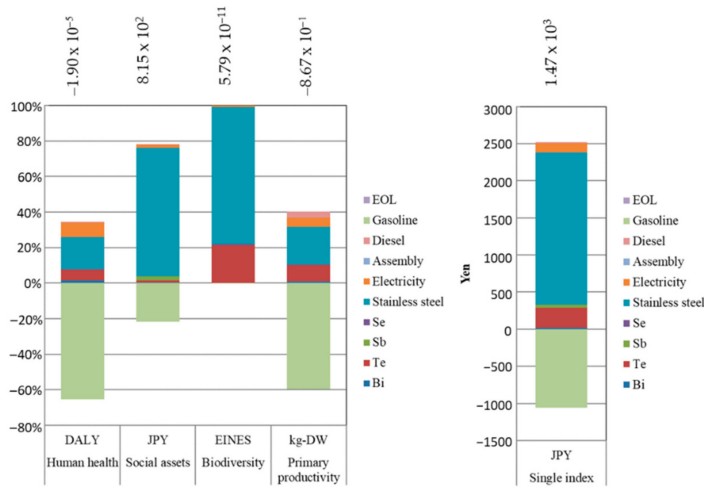

**Figure 7.** Damage and environmental cost assessment for different materials under the technology innovation scenario (Case 2).

## 4. Discussion

Our analysis showed that, with the current TEG conversion efficiency of 7.2%, deployment of automotive TEG devices does not reduce GHG emissions in Case 1 and 2. The conversion efficiency of TEG devices should therefore be improved in order to have a positive impact on reducing GHG emissions. Table 3 shows a comparison of our results with those of previous studies.

**Table 3.** Summary of previous TEG LCA results.

| Reference | TEG Type | LCA (Production) | LCA (Use) | LCA (EOL) | Impact Analysis | Scaling Effect | TEG Merit | TEG Efficiency | Application |
|---|---|---|---|---|---|---|---|---|---|
| Søndergaard at al. [24] | Organic polymer | Yes | No | No | No | No | No | No mention | Test devise |
| Ghojel. [25] | Bi-Te | Yes | Yes | Yes | Yes | No | Yes | 5% | Automobile |
| Patyk. [26] | Bi-Te | No | Yes | No | Yes | No | Yes | 10% | Automobile |
| Patyk. [27] | Bi-Te | Yes | Yes | No | Yes | No | Yes | 7.5% | Steam Expander |
| Kishita et al. [6] | Bi-Te | Yes | Yes | Yes | No | No | No | 7.2% | Automobiles |
| Irshad et al. [28] | Photovoltaic wall | No | Yes | No | No | No | Yes | 14% | Air conditioner |
| Krishnamoorthy et al. [29] | Various types | Yes | Yes | Yes | Yes | No | Yes | 10% | Coal power plant |

A variety of conclusions were reached in previous studies. Kishita et al. [6] and Søndergaard et al. [24] concluded that TEG implementation was not beneficial for reducing GHG emissions. However, Ghojel [25], Patyk [26], Patyk [27], Irshad et al. [28], and Krishnamoorthy et al. [29] concluded that TEG implementation had a positive effect on relieving environmental burdens. In order to perform a fair comparison, Bi-Te automotive TEGs were selected for the analysis. Under these conditions, only Kishita et al. [6] concluded that TEGs would not have a positive impact on the environment. Although Patyk [26] concluded that TEGs would have a positive impact on reducing GHG emissions, that study did not include any EOL analysis and used a very high conversion efficiency (10%). Ghojel [25] performed a full cradle-to-grave LCA analysis with a 5% conversion efficiency and proposed that a TEG efficiency of 5% was sustainable and beneficial for reducing GHG emissions. However, the conclusions in this analysis were completely different from that of Kishita et al. [6]. The driving distances assumed by Kishita et al. [6], Patyk [26], and Ghojel [25] were 5460 km/year, 150,000 km (lifetime), and 19,700 km/year, respectively. Kishita et al. [6] used a 12-year vehicle lifetime. So, using this lifetime, the lifetime driving distance of Kishita et al. [6], Patyk [26], and Ghojel [25] are 65,520, 150,000, and 236,400 km, respectively.

Lifetime driving distance corresponds to the use phase; the longer the lifetime driving distance, the greater the benefit of the TEG. Ghojel [25] assumed a TEG efficiency of only 5%, but the total driving distance over the lifetime of the vehicle was much longer than that of Kishita et al. [6]. Patyk [26] used a conversion efficiency of 10% with a lifetime driving distance of 150,000 km; however, these values are considered to be too optimistic. If the lifetime driving distance is short, as in Kishita et al. [6], then the analysis results would likely have been opposite to their conclusions of TEG deployment being beneficial in reducing GHG emissions. This observation implies that, in order to maximize the benefits of automotive TEG devices, the lifelong driving distance needs to be longer and/or the conversion efficiency needs to be higher, as described in the Section 3. However, considering a realistic lifetime driving distance of between 100,000 and 200,000 km [30], the assumptions of Kishita et al. [6] may be too conservative as their lifelong driving distance was short. Since our analysis is based on Kishita et al. [6], the conclusions showed that the TEG conversion efficiency needed to be improved and that the amount of stainless steel used in fabrication needed to be reduced. However, the lifetime driving distance must be considered in assessments of automotive TEG benefits. In future studies, we will include lifetime driving distance as another variable for TEG assessment.

## 5. Conclusions

This study performed a cradle-to-grave LCA analysis of an automotive TEG heat exchanger. Specifically, with a scaling effect in place, the outcomes of new technologies—from the research scale to the mass production scale—were investigated. The findings showed that direct comparisons of new technologies at the research and mass production scales are not appropriate. Rather, only when the effect of scaling is considered is it possible to assess the environmental footprint at the mass production and research scales.

As the scales of the hot press and electric furnace used to fabricate the TEG increase, electricity consumption decreases due to the scaling effect. In this study, electricity consumption remained constant after Case 3. The scaling factor for the ball mill process is >1, which implies that electricity consumption increases as the scale increases. However, the electricity consumption of the ball mill is offset by the decrease in the electricity consumption of the electric furnace and hot press. Further, the production scale of the hot press was small, which means that the hot press process could be a bottleneck for TEG production and throughput.

This study also demonstrated that, while GHG emissions were positive under the baseline scenario, they became negative ($-1.56 \times 10^2$ kg-$CO_2$ eq/kg) under the technology innovation scenario due to the GHG credits generated by the TEG in the usage phase. Some of the impacts in the impact categories were also reduced under the technology innovation

scenario. However, depending on the category, it is possible that the values of the impact categories can increase. In this study, the effect of stainless steel had a large impact on GHG emissions. In addition, Te also contributed to increasing the effect of the impact categories, particularly the ozone layer depletion and toxicity-related categories. As for the resource consumption category, Bi occupied 88% of the category, which implies that an alternative material needs to be sought for this purpose. In terms of the environmental costs (single index), the credit was equivalent to 1055 JPY, which meant that the total index was 1466 JPY. In addition, substitution of or reduction in the Te in the TEG would decrease the impacts of this element on human health, biodiversity, and primary production. Finally, in order to reduce GHG emissions, an increase in the conversion efficiency and a decrease in the stainless steel content of the TEG housing are both essential prerequisites for the commercial application of TEG heat exchangers.

However, the findings of previous studies also showed that the benefits associated with TEGs depend on the lifetime driving distance. Therefore, the payback driving distance, defined as the distance that must be driven to compensate for a GHG increase caused by TEG production by GHG savings during the use phase, should be analyzed for a given conversion efficiency, and this topic will be examined in a future study.

**Author Contributions:** Conceptualization, K.K., Y.K. and Y.S.; Methodology, K.K.; Formal analysis, K.K.; Investigation, K.K., Y.K. and Y.S.; Resources, Y.K.; Data curation, K.K.; Writing—original draft preparation, K.K.; Writing—review and editing, K.K.; Visualization, K.K.; Supervision, Y.S.; Project administration, K.K.; Funding acquisition, K.K. and Y.K. All authors have read and agreed to the published version of the manuscript.

**Funding:** Tsukuba City and the Watanabe Memorial Foundation.

**Institutional Review Board Statement:** Not applicable.

**Informed Consent Statement:** Informed consent was obtained from all subjects involved in the study.

**Data Availability Statement:** Data sharing is not applicable to this article.

**Acknowledgments:** This study was supported in part by Tsukuba City and the Watanabe Memorial Foundation. We thank Michio Kobayashi for his contributions to improving the quality of the manuscript.

**Conflicts of Interest:** The authors declare no conflict of interest.

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
