# Peer review of "Life Cycle Assessment of Thermoelectric Generators (TEGs) in an Automobile Application"

_sustainability, doi:10.3390/su132413630_

Round 1

Reviewer 1 Report

Thank you very much to tha authors that they have taken in consideration the recomendations. The reviewer thinks that the article is now more scientifically sound and also its structure helps to maintain the thread of the argument of the paper.

Author Response

Thank you for your detailed review and your acceptance of our paper. I really appreciate for your effort.

regards,

michio kobayashi

Reviewer 2 Report

The topic presented in this work is really interesting. However, several challenges are required:

I analyze the single sections:

Abstract has inappropriate structure. I suggest to answer the following aspects: - general context - novelty of the work - methodology used (describe briefly the main methods or treatments applied) - main results and related interpretations.

Introduction: This section should briefly place the study in a wide context and emphasize why it is relevant carrying out the analysis. It should define the purpose of the work and its significance. In this perspective, this section is too succinct and fails to effectively point out the relevance of your contribution towards the existing literature. For example, the authors could discuss also social aspects for LCA.

See

  • Social Life Cycle Approach as a Tool for Promoting the Market Uptake of Bio-Based Products from a Consumer Perspective 
  • Life Cycle Assessment: a review of the methodology and its application to sustainability-4

Materials and methods: I found this section very important for the readability of the paper. However, the research methodology seems underdeveloped. Methods should be described in detail. I think the research procedure could be much more clearly described by means of a diagram also highlighting its potential and limit.

Author Response

Thank you for your detailed feedback. I sincerely reply to your comment as attached.

regards,

Michio Kobayashi

Reviewer 3 Report

The authors made the improvements requested by the Reviewer.

I only suggest to improve the English before publication.

Author Response

Thank you for your review and feedback. Here is my response as attahced.

regards,

Michio Kobayashi

Round 2

Reviewer 2 Report

The paper is much improved

This manuscript is a resubmission of an earlier submission. The following is a list of the peer review reports and author responses from that submission.

Round 1

Reviewer 1 Report

Overall, the authors' work is a very interesting contribution to the scientific community. It is true that we need research that helps us to focus our efforts in a field as demanding and, at the same time, fundamental to economic and social development as energy. Therefore, the reviewer considers the investigation of the real efficiency of TEG devices by applying life cycle analysis (LCA) as a great contribution. However, the reviewer believes that by taking into account some minor revisions, the article will have a greater impact on the scientific community:
- The introduction presented by the authors can be further nourished by contributions about the different applications of life cycle analysis in comparative assertions such as the one in the reviewed paper. 
- It would be good if they could also revise the conclusion in order to make it more robust by analysing in more depth all the results obtained.
- With these two recommendations in mind, a greater number of references will be obtained, which will complete the structure of the article. 

Reviewer 2 Report

This work is really interesting and shows a good potential if several challenges are addressed:

  1. Introduction presents interesting information, but it is still poor. In fact, the reference to the existing contributions is weak and sometimes hardly updated.

Moreover, I would suggest the authors to describe also the social pillar as a general perspective.  I would appreciate to see recent scientific contribution to social life cycle assessment also with respect the cutting edge topics of sustainability (e.g. bioeconomy). See among the others:

  1. https://www.mdpi.com/2071-1050/10/2/547
  2. https://www.sciencedirect.com/science/article/pii/S2211339813000221
  3. https://www.mdpi.com/2071-1050/10/4/1031
  4. https://www.sciencedirect.com/science/article/pii/S1364032117310584

Materials and methods: I found this section very important for the readability of the paper. The research methodology seems underdeveloped. Methods should be described in detail. I think the research procedure could be much more clearly described by means of a diagram also highlighting its potential and limit. 

Discussions: The discussion of the results is merely descriptive and the obtained evidence is flimsy due to the fact the outcomes are not supported by an adequate discussion in light of scientific literature. Authors should discuss the results and how they can be interpreted in perspective of previous studies and their implications should be discussed in the broadest context possible. 

Conclusions: Conclusions must also be revised according to the previous comments. In particular, they should discuss practical and policy implications as well as future lines of research. Is there space for financial implications (e.g. green finance) to increase conversion efficiency, etc..?

Please see:

http://www.inderscience.com/offer.php?id=109735

Reviewer 3 Report

The proposed research look interesting, on the other hand the novelties are not well specified and the methodology needs to be much improved.

In particular:

-The literature review needs to be improved and discussed by emphasising the current drawbacks.

To provide an example consider this review.

Tsoy, N., Steubing, B., van der Giesen, C., & Guinée, J. (2020). Upscaling methods used in ex ante life cycle assessment of emerging technologies: a review. The International Journal of Life Cycle Assessment, 1-13.

-Novelty of the proposed work need to be better explained. It seems that the novelty only concerns the application of a LCA to a specific a case study. This is not enough to consider the paper as an “Original research manuscripts”.

-Methodology need to be rewritten and differentiated from the case study.

In particular:

Section 2.1. Functional units and system boundaries.

It is not clear the meaning of this sub-section and why it is in the first paragraph of the methodology section. This it is not methodology but the description of the case study.

Section 2.2. Inventory analysis and 2.3. Scenario analysis

Also in this sub-sections, it is not clear what is methodology and what is the application of the methodology the specific case study.

-Conclusion

The obtained results are interesting, but it is not clear how the proposed research and the achieved results can be useful (from a theoretical and practical point of view) for engineers, technicians and other stakeholders.

how do the results improve the sector literature? why are they important?

In conclusion I believe that the work has the potential to be published. On the other hand, the manuscript needs to be deeply reworked by better specifying novelty, importance of the research in the literature, methodology (differentiating from the case study), and importance of the results.

Minor comment:

- “GHG” has not been defined in the abstract.